# Formin-like 1 (FMNL1) Is Associated with Glioblastoma Multiforme Mesenchymal Subtype and Independently Predicts Poor Prognosis

**DOI:** 10.3390/ijms20246355

**Published:** 2019-12-17

**Authors:** Nayuta Higa, Yoshinari Shinsato, Muhammad Kamil, Takuro Hirano, Tomoko Takajo, Michiko Shimokawa, Kentaro Minami, Masatatsu Yamamoto, Kohichi Kawahara, Hajime Yonezawa, Hirofumi Hirano, Tatsuhiko Furukawa, Koji Yoshimoto, Kazunori Arita

**Affiliations:** 1Department of Neurosurgery, Graduate School of Medical and Dental Sciences, Kagoshima University, Kagoshima 890-8520, Japanhajime@m3.kufm.kagoshima-u.ac.jp (H.Y.); hirahira@m2.kufm.kagoshima-u.ac.jp (H.H.); kyoshimo@m.kufm.kagoshima-u.ac.jp (K.Y.);; 2Department of Molecular Oncology, Graduate School of Medical and Dental Sciences, Kagoshima University, Kagoshima 890-8544, Japan; yosinari@m3.kufm.kagoshima-u.ac.jp (Y.S.); hira-2323@wb3.so-net.ne.jp (T.H.); k3340499@kadai.jp (M.S.); kminami@m2.kufm.kagoshima-u.ac.jp (K.M.); masatatu@m2.kufm.kagoshima-u.ac.jp (M.Y.); k-kawahr@m3.kufm.kagoshima-u.ac.jp (K.K.); 3Department of Neurosurgery, Faculty of Medicine, Airlangga University, Surabaya 60132, Indonesia; 4Department of Digestive Surgery, Breast and Thyroid Surgery, Graduate School of Medical and Dental Sciences, Kagoshima University, Kagoshima 890-8520, Japan; 5Center for the Research of Advanced Diagnosis and Therapy of Cancer, Graduate School of Medical and Dental Sciences, Kagoshima University, Kagoshima 890-8544, Japan

**Keywords:** formin-like 1 (FMNL1), DIAPH1, GOLGA2, invasion, migration, mesenchymal subtype

## Abstract

Glioblastoma multiforme (GBM), the most common primary malignant brain tumor in adults, is characterized by rapid proliferation, aggressive migration, and invasion into normal brain tissue. Formin proteins have been implicated in these processes. However, the role of formin-like 1 (FMNL1) in cancer remains unclear. We studied FMNL1 expression in glioblastoma samples using immunohistochemistry. We sought to analyze the correlation between FMNL1 expression, clinicopathologic variables, and patient survival. Migration and invasion assays were used to verify the effect of FMNL1 on glioblastoma cell lines. Microarray data were downloaded from The Cancer Genome Atlas and analyzed using gene set enrichment analysis (GSEA). FMNL1 was an independent predictor of poor prognosis in a cohort of 217 glioblastoma multiforme cases (*p* < 0.001). FMNL1 expression was significantly higher in the mesenchymal subtype. FMNL1 upregulation and downregulation were associated with mesenchymal and proneural markers in the GSEA, respectively. These data highlight the important role of FMNL1 in the neural-to-mesenchymal transition. Conversely, FMNL1 downregulation suppressed glioblastoma multiforme cell migration and invasion via DIAPH1 and GOLGA2, respectively. FMNL1 downregulation also suppressed actin fiber assembly, induced morphological changes, and diminished filamentous actin. FMNL1 is a promising therapeutic target and a useful biomarker for GBM progression.

## 1. Introduction

Glioblastoma multiforme (GBM) is the most common primary malignant brain tumor in adults and is classified as grade IV by the World Health Organization (WHO). The GBM relapse rate is very high, and median survival is typically only 10 to 11 months, even when multimodal treatment, encompassing surgery, radiation, and chemotherapy, is applied [1,2]. Rapid proliferation, aggressive migration, and invasion into surrounding healthy brain tissue, and even into the contralateral hemisphere, are characteristic of GBM and prevent total surgical cure [3]. Invasive GBM cells can escape irradiation. Hence, the development of innovative therapies against the migratory and invasive tumor cells may significantly improve clinical outcomes.

Cytoskeletal remodeling processes, which are related to cell polarization, adhesion, division, and migration, are strongly dependent on formin proteins function [4]. Formin proteins have been implicated in cancer progression, invasion, and migration. In mammals, 15 formin proteins have been identified and grouped into seven subclasses: diaphanous proteins (DIA), formin homology domain-containing proteins (FHOD), inverted formin (INF), formins (FMN), formin-related proteins in leukocytes (FMNLs/FRLs), disheveled-associated domain proteins, and delphilin [5]. Formin-related proteins in leukocytes include FMNL1 (FRL1), FMNL2 (FRL3), and FMNL3 (FRL2), all of which have similar domain organization as the diaphanous proteins [6]. Several formins stimulate actin polymerization and enhance cell motility. For example, FMNL1 mediates the assembly of filamentous actin networks and organizes actin filaments into bundles. FMNL1 also stabilizes the Golgi complex and regulates phagocytosis, cell adhesion, podosome dynamics, cell migration, and survival in macrophages [7,8]. FMNL1 is naturally enriched in hematopoietic cells and tissues such as thymus, spleen, and peripheral blood leukocytes. It is overexpressed in malignant lymphomas in patients with chronic lymphocytic leukemia, as well as in T cells in patients with malignant non-Hodgkin’s lymphoma [9,10]. Although the role of FMNL1 in GBM is unclear [11], formin-related proteins have been implicated in the pathophysiology of several other tumors. Thus, FMNL1 mediates nasopharyngeal carcinoma cell aggressiveness [12]. FMNL2 is implicated in the invasion of colorectal cancer and melanoma [13,14,15]. FMNL3 promotes migration, proliferation, and metastasis in colorectal cancer, melanoma, esophageal squamous cell carcinoma, and nasopharyngeal carcinoma [16,17,18]. As a result, FMNL1, FMNL2, and FMNL3 are correlated with poor prognosis in several cancers [12,13,14,15,16,17,18].

Here, we report that FMNL1 expression is associated with unfavorable prognosis in GBM and that FMNL1 regulates tumor migration and invasion via DIAPH1 and GOLGA2. DIAPH1 was previously identified as a possible FMNL1 downstream effector that regulates actin assembly during meiosis [19]. Meanwhile, GOLGA2 was previously found to colocalize with FMNL1 and is required to stabilize the Golgi complex [6]. We examined the role of DIAPH1 and GOLGA2 in FMNL1 promotion of cell motility. Our data indicate that FMNL1 expression is associated with GBM migration and invasion by regulating DIAPH1 and GOLGA2. FMNL1 is involved in the GBM malignant phenotype.

## 2. Results

### 2.1. Relationship between FMNL1 Expression and Clinicopathology

Variable FMNL1 expression was detected in all 217 GBM surgical patients’ specimens, but not in normal brain tissue (Figure 1A–C and Appendix A). FMNL1 levels were higher in recurrent than in primary tumors (Appendix A). Based on quantification using color deconvolution with Image J software (National Institutes of Health, Bethesda, MD, USA), FMNL1 expression was modest and strong in 49.8% (108/217) and 50.2% (109/217) of patients, respectively. FMNL1 levels were associated with age (*p* = 0.006), Karnofsky performance status (KPS; *p* = 0.039), extent of surgical resection (*p* = 0.01), and number of surgeries (*p* < 0.001).

### 2.2. Relationship between FMNL1 Expression and Prognosis

Overall, survival was significantly shorter in patients strongly expressing FMNL1 than in patients modestly expressing FMNL1, both in our cohort (*p* < 0.001, Figure 1D) and in the TCGA GBM cohort (*p* = 0.017, Appendix A).

High FMNL1 gene expression was associated with the mesenchymal GBM subtype (Figure 1E). VERHAAK and PHILLIPS glioblastoma mesenchymal genes [20,21] and VERHAAK and PHILLIPS glioblastoma proneural genes [20,21] were significantly associated with high and low FMNL1 expression, respectively (Figure 1F and Appendix A). However, VERHAAK glioblastoma classical and neural genes [21] were not associated with FMNL1 expression (Appendix A). These results were consistent with the correlation between high FMNL1 expression and poor prognosis in our cohort. The mesenchymal glioma subtype has been associated with worse prognosis in comparison to the proneural glioma subtype [22,23].

To evaluate the relationship between FMNL1 expression and the mesenchymal GBM subtype, we quantified the expression levels of six genes we had previously profiled as representative mesenchymal markers (CHI3L1, CD44, VIM, RELB, TRADD, and PDPN) [24]. These mesenchymal markers were significantly upregulated in cells overexpressing FMNL1 compared to cells overexpressing enhanced green fluorescence protein (EGFP) (Appendix A). The FMNL1 knockdown effect on gene expression was limited, possibly due to insufficient siRNA-induced transient suppression (Appendix A). These results suggested that FMNL1 expression not only regulated downstream targets but may also modulate GBM mesenchymal-subtype-associated genes.

### 2.3. Univariate and Multivariate Survival Analyses

Univariate analysis showed that age (*p* < 0.001), KPS (*p* < 0.001), extent of surgical resection (*p* < 0.001), number of surgeries (*p* = 0.009), chemotherapy (*p* < 0.001), radiation therapy (*p* < 0.001), and FMNL1 expression (*p* < 0.001) were significant predictors of postoperative survival. Multivariate analysis of 10 parameters revealed that age (*p* = 0.009), extent of surgical resection (*p* = 0.004), chemotherapy (*p* < 0.001), radiation therapy (*p* < 0.001), and FMNL1 expression (*p* < 0.001) were independent prognostic factors (Table 1).

### 2.4. FMNL1 Knockdown Reduces GBM Migration and Invasion

U251MG and DBTRG-05MG cells highly expressed endogenous FMNL1. FMNL1 expression was downregulated significantly in these cells transfected with any of the three siRNAs designed against FMNL1, but not in the cells transfected with control siRNA (Figure 2A,B). FMNL1 knockdown decreased the number of migratory and invasive cells (Figure 2C,D), as well as the matrix metalloproteinase 9 (MMP9) activity in conditioned medium determined by gelatin zymography analysis (Figure 2E). The FMNL1 effect knockdown on cell proliferation was not significant during the study period per the invasion and migration assays (Appendix A).

### 2.5. FMNL1-Mediated GBM Migration Depends on DIAPH1

DIAPH1 was downregulated significantly following FMNL1 knockdown per Western blot analysis (Figure 3A), following a previous study suggesting FMNL1 controls DIAPH1 expression to regulate actin assembly during meiosis [19]. DIAPH1 knockdown by siRNA, without changing FMNL1, as confirmed by Western blot (Figure 3B), inhibited U251MG and DBTRG-05MG cell migration but not invasion (Figure 3C). These results implied that FMNL1 promotes GBM migration via DIAPH1.

### 2.6. FMNL1-Mediated GBM Invasion Depends on GOLGA2

Western blot analysis showed that GOLGA2 was downregulated significantly after FMNL1 knockdown (Figure 3D), following a previous study suggesting that FMNL1 regulates GOLGA2 distribution and expression during spindle formation in mouse oocytes [6]. GOLGA2 knockdown by siRNA, without changing FMNL1 as confirmed by Western blotting (Figure 3E), significantly inhibited invasion but only modestly suppressed migration (Figure 3F). Altogether, the data indicated that FMNL1 mediates GBM invasion via GOLGA2.

### 2.7. FMNL1 Overexpression Induces Migration but Not Invasion

U87MG and KNS81 cell lines expressed only low levels of FMNL1. We established stably expressing FLAG-tagged FMNL1 or FLAG-tagged EGFP from these cell lines by infection with corresponding lentiviral vectors. FMNL1 expression was higher (Figure 4A) and migration was significantly enhanced (Figure 4B) in cells infected with FMNL1 lentiviral expression vector, as compared to cells infected with EGFP lentiviral expression vector. The invasion was comparable between the cell lines (Figure 4B). DIAPH1 was significantly more upregulated in cells overexpressing FMNL1 than in cells overexpressing EGFP (Figure 4C), as assessed by Western blot, although GOLGA2 expression was comparable between the cells (Figure 4D). Collectively, the data indicated that FMNL1 regulates GBM migration and invasion via DIAPH1 and GOLGA2, respectively.

### 2.8. FMNL1 Knockdown Compromises the Actin Cytoskeleton via DIAPH1 but not GOLGA2

In GBM cells transfected with siRNA against FMNL1, structured actin fibers were lost, morphological changes were observed, and filamentous actin significantly diminished (Figure 5A,B), highlighting FMNL1 as a regulator of actin assembly. Similar results were obtained in cells transfected with siRNA targeting DIAPH1 but not in cells transfected with siRNA against GOLGA2 (Figure 5A,B). These findings suggested that FMNL1 regulates the actin cytoskeleton in GBM cells via DIAPH1 but not GOLGA2.

### 2.9. Identification of FMNL1-Associated Biological Pathways

The GSEA of GBM cases in TCGA identified the gene sets GO_LAMELLIPODIUM, KEGG_FOCAL_ADHESION, GO_INVADOPODIUM, BUDHU_LIVER_CANCER_METASTASIS_UP, CROMER_METASTASIS_UP, CELL_MIGRATION, WU_CELL_MIGRATION, GO_ACTIN_CYTOSKELETON, and GO_ACTIN_FILAMENT_POLYMERIZATION as being significantly associated with FMNL1 expression (Appendix A). These results were consistent with our findings concerning the mesenchymal glioma phenotype.

## 3. Discussion

Given the lack of reliable prognostic biomarkers and therapeutic targets, GBM treatment remains challenging, and patient survival is poor. Thus, identifying key molecules mediating GBM migration and invasion is crucial. Actin-based processes, including cell polarization, cell division, membrane trafficking, migration, morphogenesis, and filopodium formation [25], are increasingly being implicated in mediating advanced tumor invasion into adjacent tissues and promoting metastases. Formins, which are potent regulators of actin dynamics, may be involved in tumor invasion and metastasis promotion; and emerging evidence has already implicated FMNL1, FMNL2, and FMNL3 in these processes [12,13,14,15,16,17,18]. However, the role of FMNL1 in GBM remains unclear. To the best of our knowledge, this is the first study to investigate FMNL1 in GBM.

We detected variable FMNL1 expression levels in GBM tissues but not in normal brain tissues. FMNL1 was previously not detected in normal neural parenchymal cells [26]. We found that high FMNL1 expression is an independent predictor of unfavorable GBM prognosis. Age, KPS, the extent of surgical resection, chemotherapy, and radiation therapy were also identified by multivariate analysis as independent GBM prognostic factors, in line with many previous reports [27,28,29], highlighting the robustness of our study. Notably, FMNL1 expression was significantly associated with age, KPS, and the extent of surgical resection. The association with KPS and the number of surgeries suggested that mesenchymal subtypes with FMNL1 expression were related to invasive phenotype affecting patients’ status. (Appendix A). However, poor patient status, associated with an immunological condition and GBM microenvironment, may have affected the invasive tumor phenotype, including FMN1 expression. GSEA confirmed that high and low FMNL1 expressions are associated with the mesenchymal and proneural signature, respectively. This follows other studies demonstrating progressing gliomas acquiring mesenchymal features and losing proneural features. Mesenchymal transition is associated with tumor aggressiveness, therapy resistance, and poor clinical outcome [30]. Thus, our clinical findings were consistent with the literature and supported the hypothesis that FMNL1 is associated with and plays essential roles in GBM mesenchymal transition.

In GBM cells, FMNL1 knockdown suppressed both migration and invasion, highlighting its potential role in GBM malignancy. The significant association of FMNL1 with lamellipodia, focal adhesion, invadopodia, metastasis, and migration-related gene sets further reinforces the notion of FMNL1 involvement in GBM. In contrast, FMNL1 overexpression enhanced cell migration but not invasion. DIAPH1 and GOLGA2 [9,19], possible FMNL1 downstream targets, were significantly diminished after FMNL1 knockdown, but only DIAPH1 was upregulated in cells overexpressing FMNL1. The basis for this discrepancy is unknown. One possibility is a shortage of other vital proteins working together with FMNL1. FMNL1 expression alone is not enough to enhance cell-line-dependent invasion [31,32]. Another option is a cumulative effect of at least 11 FMNL1 splice variants and 10 protein isoforms. Thus, siRNAs against FMNL1 may knock down several splice variants, including the full-length transcript, whereas only the full-length protein was overexpressed exogenously. We note that alternative splicing is a crucial mechanism generating various proteins with different, even antagonistic, biological functions [33,34].

DIAPH1 is a multidomain protein that, like mDia2 and mDia3, is a diaphanous-related formin. By facilitating cytoskeletal rearrangements, DIAPH1 mediates various biological processes. This includes morphogenesis, cytokinesis, cell polarization, adhesion, and migration [35,36,37,38]. DIAPH1 downregulation was reported to suppress invasion and/or migration in breast cancer, colon cancer, and glioblastoma [39,40,41]. We found that FMNL1 regulates DIAPH1 expression not only to control actin assembly during meiosis as previously described [19] but also to promote GBM cell migration. However, GOLGA2 is a cis-Golgi protein that stabilizes the Golgi apparatus [42,43,44], probably in coordination with FMNL1. GOLGA2 and FMNL1 colocalization have been demonstrated in HeLa cells [7]. FMNL1 was reported to regulate GOLGA2 distribution and expression during spindle formation in mouse oocytes [6], whereas GOLGA2 downregulation suppressed gastric and lung cancers invasion [44,45]. Similarly, we demonstrated that FMNL1 plays an essential role in cell invasion and GOLGA2 expression in GBM cells. The precise function of GOLGA2 in invasion and migration remains unclear and we are planning to study its downstream molecules, including MMPs.

Notably, FMNL1 knockdown diminishes actin polymerization and alters GBM cell morphology, consistent with its ability to regulate actin at various locations and to associate with membrane structures. GSEA of GBM cases from TCGA corroborates this result. FMNL1 knockdown was reported to increase filamentous actin in HeLa and Jurkat T cells [7]. This suggests the role of FMNL1 in regulating filamentous actin and cell morphology may be cell- or tissue-type-dependent.

FMNL1 is predominantly expressed in hematopoietic cells. It has several specific functions like microtubule-organizing center reorientation in T cells and podosomes activation in macrophages. Overall, FMNL1 has common functions with FMNL2/3. However, little is known about the functional differences between FMNL1 and FMNL2. FMNL1 ectopic expression in GBM may affect gliomagenesis since only FMNL1 has been reported to interact with AKT [10].

Our results showed that high FMNL1 expression in GBM patients is correlated with an unfavorable prognosis. We demonstrated that FMNL1 regulates GBM cell migration via DIAPH1 and plays an important role in invasion via GOLGA2. These results suggested that FMNL1 is a useful biomarker and a promising therapeutic target for GBM.

## 4. Materials and Methods

### 4.1. Reagents and Antibodies

RPMI 1640 and fetal bovine serum (FBS) were purchased from Nikken Biomedical Laboratory (Osaka, Japan) and PAA Laboratories (Mississauga, ON, Canada), respectively. MTT (3-(4,5-dimethylthiazol-2-yl)-2,5-diphenyl tetrazolium bromide), monoclonal antibodies against FLAG M2 (Cat#F1804), and polyclonal antibodies against FMNL1 (HPA008129) were obtained from Sigma-Aldrich (St. Louis, MO, USA). DAPI (4′,6-diamidino-2-phenylindole dihydrochloride) and ActinRed^TM^ 555 ReadyProbes^TM^ were purchased from Dojindo Laboratories (Kumamoto, Japan) and Life Technologies (Melbourne, Australia), respectively. Monoclonal antibodies against GAPDH were procured from Cell Signaling Technology (Danvers, MA, USA), whereas monoclonal antibodies against DIAPH1 (Cat#610849) and GOLGA2 (Cat#610823) were obtained from BD Biosciences (San Jose, CA, USA).

### 4.2. Patients and Tumor Samples

Clinical specimens were obtained from surgically removed and pathologically confirmed as GBM tumors. They originated from 217 GBM patients treated from 2000 to 2015 at the Department of Neurosurgery, Kagoshima University Hospital (Kagoshima, Japan). The study was approved by the Institutional Review Board of Kagoshima University and complied with the Helsinki Declaration. Informed consent was obtained from each patient.

### 4.3. Immunohistochemistry

Surgical specimens were fixed within 10 min of excision in 10% neutral buffered formaldehyde for 24 h, embedded in paraffin, sectioned at 3 μm, and mounted on glass slides coated with poly-l-lysine. Subsequently, sections were probed with 1:100 anti-FMNL1 antibody and stained with diaminobenzidine tetrahydrochloride and hematoxylin. Immunoreactivity was quantified in four fields observed at 200× magnification. This was done using color deconvolution in Image J software (National Institutes of Health, Bethesda, MD, USA), as described by Hirano et al. [46].

### 4.4. Cell Culture

Human glioblastoma cell lines U251MG (Japanese Collection of Research Bioresources Cell Bank, Osaka, Japan), DBTRG-05MG (American Type Culture Collection, Manassas, VA, USA), U87MG (Health Protection Agency Culture Collections, Salisbury, U.K.), and KNS81 (Japanese Collection of Research Bioresources Cell Bank, Osaka, Japan) were routinely cultured at 37 °C in a humidified atmosphere with 5% CO_2_ in RPMI 1640 medium supplemented with 10% FBS and 100 U/mL penicillin.

### 4.5. siRNA Transfection

We designed siRNAs FMNL1 #1 (sense strand 5′-CGUCUGUAUUAUGUGCCUA-3′ and anti-sense strand 5′-UAGGCACAUAAUACAGACG-3′), FMNL1 #2 (sense strand 5′-CCUGGUGAAGGUCAUUGCU-3′ and anti-sense strand 5′-AGCAAUGACCUUCACCAGG-3′), and FMNL1 #3 (sense strand 5′-UCCGCUGUGGCCCGCCUCAAA-3′ and anti-sense strand 5′-UUUGAGGCGGGCCACAGCGGA-3′). siRNAs specific for DIAPH1 (sc-35190) and GOLGA2 (sc-41224) were obtained from Santa Cruz Biotechnology (Santa Cruz, CA, USA). Negative control siRNA (AM4611) was purchased from Ambion (Austin, TX, USA). Following the manufacturer’s instructions, 50 nM siRNA was transfected into U251MG and DBTRG-05MG cells for 48 h at 37 °C, using RNAiMAX transfection reagent (Thermo Fisher Scientific, Carlsbad, CA, USA). The cells were then washed with PBS and cultured in RPMI 1640.

### 4.6. FMNL1 Lentiviral Expression Vector

Full-length human FMNL1 was amplified from U251MG cDNA and inserted as a FLAG-tagged fragment into the lentivirus vector CSII-CMV-MCS-IRES2-Bsd. Recombinant lentiviruses were then produced in 293FT cells via transient cotransfection with the packaging plasmids pMDLg/pRRE, pRSV-REV, and pMD2.G (Addgene, Cambridge, MA, USA) using Lipofectamine 2000 (Thermo Fisher Scientific). The obtained lentiviruses were used to infect the U87MG and KNS81 cells for 48 h. Infected cultures were then incubated with blasticidin S hydrochloride (Kaken Pharmaceutical, Tokyo, Japan), for at least five days. FLAG-FMNL1 expression in the cultures was confirmed by Western blotting; however, no clones were purified.

### 4.7. Transwell Migration and Invasion Assay

To assess invasion, 3 × 10^4^–1.5 × 10^5^ cells (depending on the cell line) were plated with serum-free medium into the top chamber of a 24-well BioCoat^TM^ Matrigel Invasion Chamber with 8 µm pores (Corning Inc., Corning, NY, USA). The bottom chamber was filled with medium containing 10% FBS. Migration was assessed similarly to invasion; however, BioCoat^TM^ Control Inserts with 8-µm pores (Corning Inc.) were used. Chamber membranes were coated with 10 µg/mL fibronectin (Sigma-Aldrich, St. Louis, MO, USA), especially in assays with U87MG cells. After 24 h in culture, the bottom membranes were fixed in 4% paraformaldehyde and stained with hematoxylin. Cells were quantified in five random fields at 200× magnification.

### 4.8. Cell Proliferation Assay

Equal numbers of cells (5 × 10^2^) were inoculated into each well and incubated for 1, 2, 3, 4, and 5 days. Cell viability was measured using the MTT colorimetric assay.

### 4.9. Confocal Laser Microscopy

Cells were fixed for 10 min with 4% paraformaldehyde in PBS, permeabilized with 0.3% Triton X-100 in PBS, blocked with 5% bovine serum albumin in PBS, and stained with DAPI and ActinRed^TM^ 555 ReadyProbes^TM^ (Life Technologies, Gaithersburg, MD, USA). They were imaged on a Zeiss LSM 700 confocal laser microscope (Zeiss, Oberkochen, Germany).

### 4.10. Quantification of Filamentous and Globular Actin

Filamentous and globular actin were assayed in U251MG and DBTRG-05MG cells with the G-actin/F-actin In vivo Assay Kit (Cytoskeleton, Denver, CO, USA), as described in the manufacturer’s online data supplement.

### 4.11. RNA Isolation and cDNA Synthesis

Total RNA was isolated from cultured cells using TRIzol (Thermo Fisher Scientific), and reverse transcribed using the ReverTra Ace Kit (Toyobo, Osaka, Japan) according to the manufacturer’s instructions [47].

### 4.12. RT-PCR

FMNL1 mRNA was quantified by Step One Plus™ (Applied Biosystems, Foster City, CA, USA) RT-PCR, using Go Taq qPCR Master Mix (Promega, Madison, WI, USA) according to the manufacturer’s instructions. Target gene expression was quantified with the comparative cycle threshold method, using GAPDH as the control. Forward and reverse primer sequences are shown in Appendix A.

### 4.13. Protein Extraction and Western Blotting

Whole-cell lysates were obtained with radio-immunoprecipitation assay buffer. This consisted of 25 mM Tris-HCl pH 7.5, 150 mM NaCl, 1% Nonidet *p*-40, 0.1% sodium dodecyl sulfate (SDS), 0.5% sodium deoxycholate, and proteinase inhibitor cocktail (Nacalai Tesque, Kyoto, Japan). Total lysate proteins were quantified using a BioRad protein assay kit (Hercules, CA, USA). Equal amounts of protein (50 µg) were separated by electrophoresis on 5–20% SDS-polyacrylamide gel (ATTO, Tokyo, Japan). Then, they were transferred onto polyvinylidene difluoride membranes (EMD Millipore, Billerica, MA, USA). The membranes were then blotted overnight at 4 °C with 1:1000 anti-FMNL1, 1:1000 anti-DIAPH1, 1:250 anti-GOLGA2, or 1:300 anti-GAPDH antibody, and visualized as described previously in Reference [48].

### 4.14. In Silico Analyses

Raw gene expression data collected from GBM tissues using Affymetrix Human U133A microarrays were downloaded from The Cancer Genome Atlas (TCGA), log_2_ transformed and normalized using the robust multi-array average (RMA). Specimens were categorized into clinical and molecular subclasses, according to a previous report [49]. Gene set enrichment was analyzed against MSigDB using the online tools provided by the Broad Institute (http://www.broadinstitute.org/gsea/index.jsp). For each analysis, 1000 gene set permutations were performed. The nominal *p*-value, false discovery rate (FDR), and normalized enrichment score (NES) were used to sort the enriched pathways in each phenotype. *p* < 0.05 and FDR < 0.25 were considered statistically significant.

### 4.15. Gelatin Zymography Assay

Cells were plated in a 6-cm dish and incubated in RPMI 1640 medium supplemented with 10% FBS for 24 h. After washing with serum-free RPMI 1640 medium, the cells were cultured in the conditioned medium for 48 h and subjected to electrophoresis. The gelatin zymography assay was performed using a Gelatin Zymo-Electrophoresis Kit (Primary Cell, Sapporo, Japan) according to the manufacturer’s directions.

### 4.16. Statistical Analysis

Data were analyzed in EZR (Saitama Medical Center, Jichi Medical University, Saitama, Japan), a graphical user interface for R (The R Foundation for Statistical Computing, Vienna, Austria). Groups were compared using the *χ*^2^ test, Student’s *t*-test, and analysis of variance (ANOVA). Patients were stratified based on median FMNL1 expression and compared by the log-rank test of Kaplan–Meier survival curves. Univariate and multivariate Cox regression analyses were performed. Differences were considered significant at *p* < 0.05.

## 5. Conclusions

Our results showed that high FMNL1 expression in GBM patients is correlated with an unfavorable prognosis. FMNL1 downregulation suppressed GBM cell migration and invasion via DIAPH1 and GOLGA2, respectively. FMNL1 downregulation also suppressed actin fiber assembly, induced morphological changes, and diminished filamentous actin. FMNL1 is a promising therapeutic target and a useful biomarker for GBM progression.

## Figures and Tables

**Figure 1 ijms-20-06355-f001:**
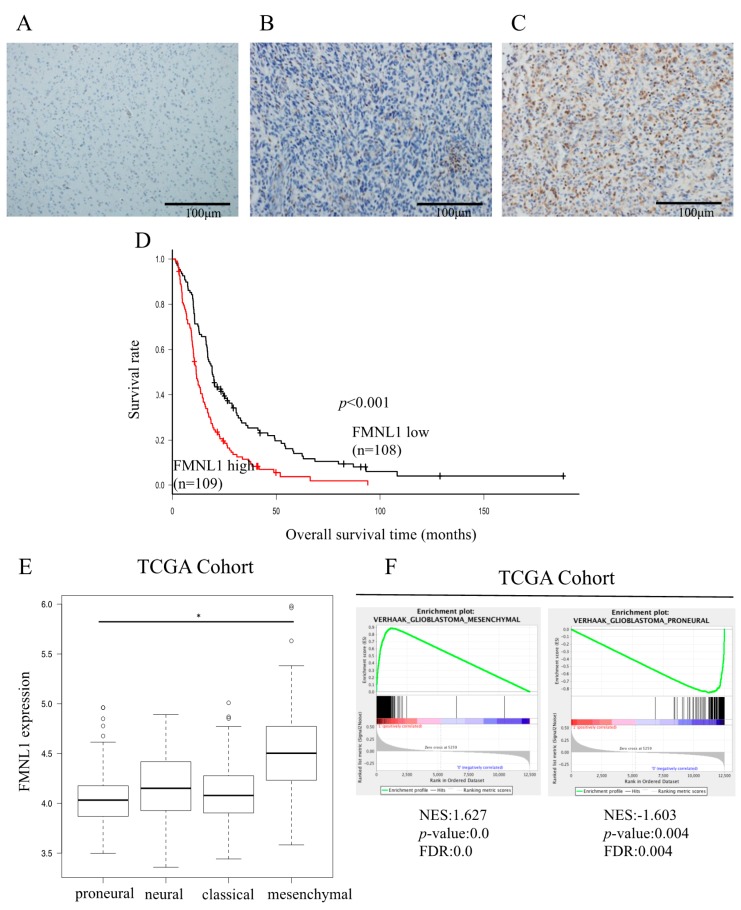
FMNL1 expression is associated with poor glioblastoma multiforme (GBM) prognosis. (**A**–**C**) FMNL1 was not detected by immunohistochemistry in (**A**) normal tissues but maybe (**B**) modestly or (**C**) strongly expressed in GBM tissues. Original magnification 200×. (**D**) Overall survival is significantly lower in patients with high FMNL1 expression than in those with modest FMNL1 expression (*p* = 1.34 × 10^−5^). Survival rates were calculated using the Kaplan–Meier method and compared using the log-rank test. (**E**) FMNL1 expression levels in the mesenchymal GBM subtype were higher than in the proneural GBM subtype per the ANOVA with the Holm test, with bars indicating SD. * *p* < 0.01. (**F**) Gene set enrichment plots for VERHAAK_GLIOBLASTOM A_MESENCHYMAL and VERHAAK_GLIOBLASTOMA_PRONEURAL in GBM specimens from TCGA.

**Figure 2 ijms-20-06355-f002:**
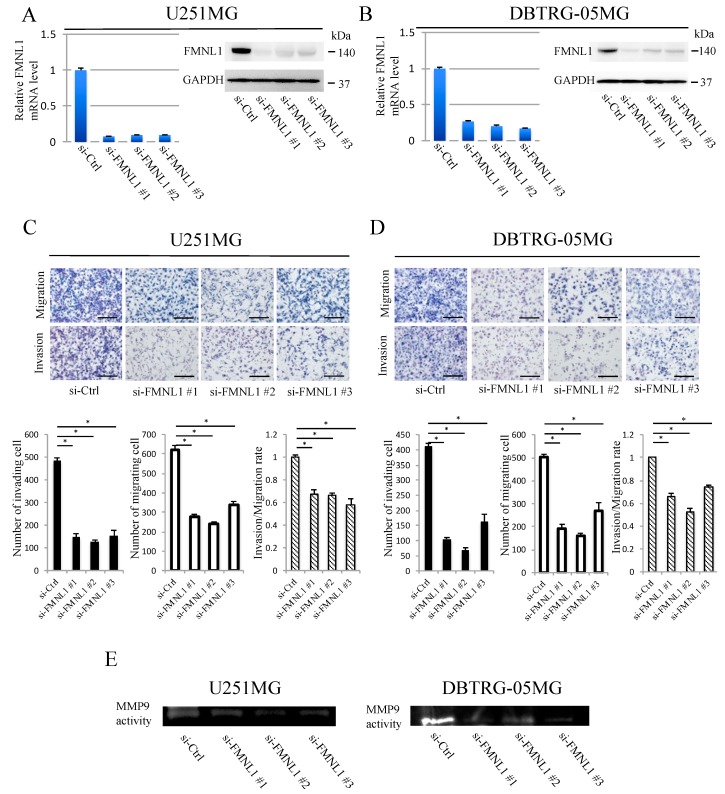
FMNL1 knockdown suppresses GBM migration and invasion. (**A**,**B**) Reverse transcription (RT)-PCR and Western blot of FMNL1 in (**A**) U251MG and (**B**) DBTRG-05MG cells transfected with control or FMNL1 siRNAs. GAPDH was used as a loading control in the Western blot assays. (**C**,**D**) FMNL1 knockdown suppressed migration and invasion of (**C**) U251MG and (**D**) DBTRG-05MG cells. Representative images of transwell migration and invasion of FMNL1 knockdown cells at 200× magnification that were quantified. Scale bar: 100 μm. Columns represent migrating or invading cells from five independent microscopic fields, with bars indicating SD. * *p* < 0.01 vs. cells transfected with control siRNA. FMNL1 knockdown suppressed MMP9 activity; (**E**) MMP9 activity in the cell-conditioned medium was evaluated using gelatin zymography.

**Figure 3 ijms-20-06355-f003:**
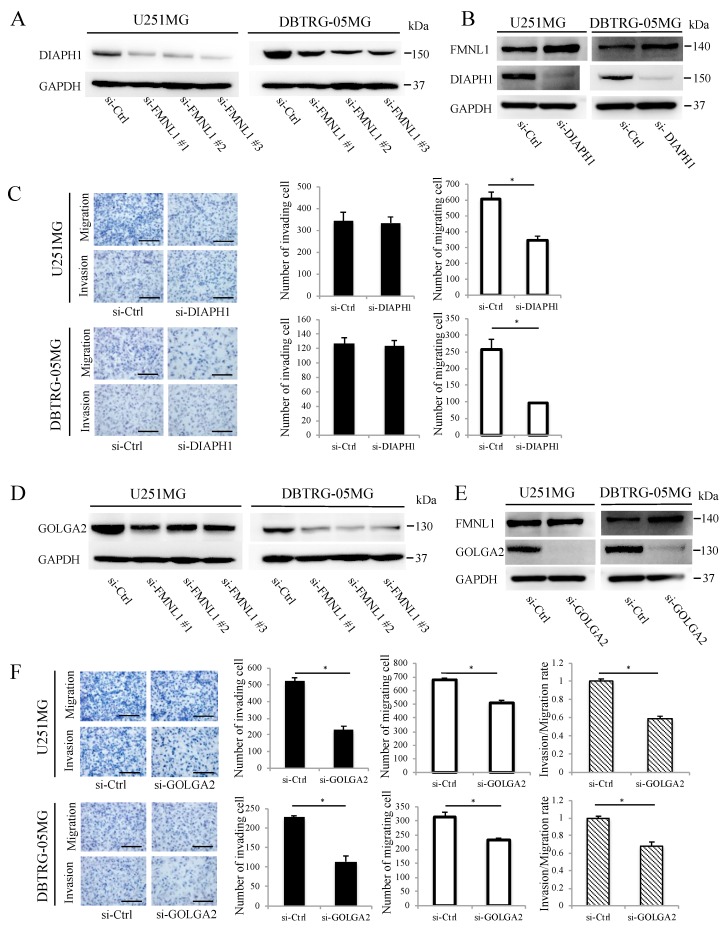
FMNL1-mediated GBM migration and invasion depend on DIAPH1 and GOLGA2, respectively. (**A**,**B**) DIAPH1 and FMNL1 protein expression in the cells transfected with siRNA against (**A**) FMNL1 and (**B**) DIAPH1. (**C**) Representative images and quantification of migration and invasion of DIAPH1 knockdown cells. Original magnification 200×; scale bar: 100 μm. (**D**,**E**) Protein expression of GOLGA2 and FMNL1 in the cells transfected with siRNAs against (**D**) FMNL1 and (**E**) GOLGA2, with GAPDH as a loading control. (**F**) Migration and invasion of GOLGA2 knockdown cells, which were quantified. Original magnification 200×; scale bar: 100 μm. Columns represent migrating or invading cells from five independent microscopic fields, with bars indicating SD. * *p* < 0.01 vs. cells transfected with control siRNA.

**Figure 4 ijms-20-06355-f004:**
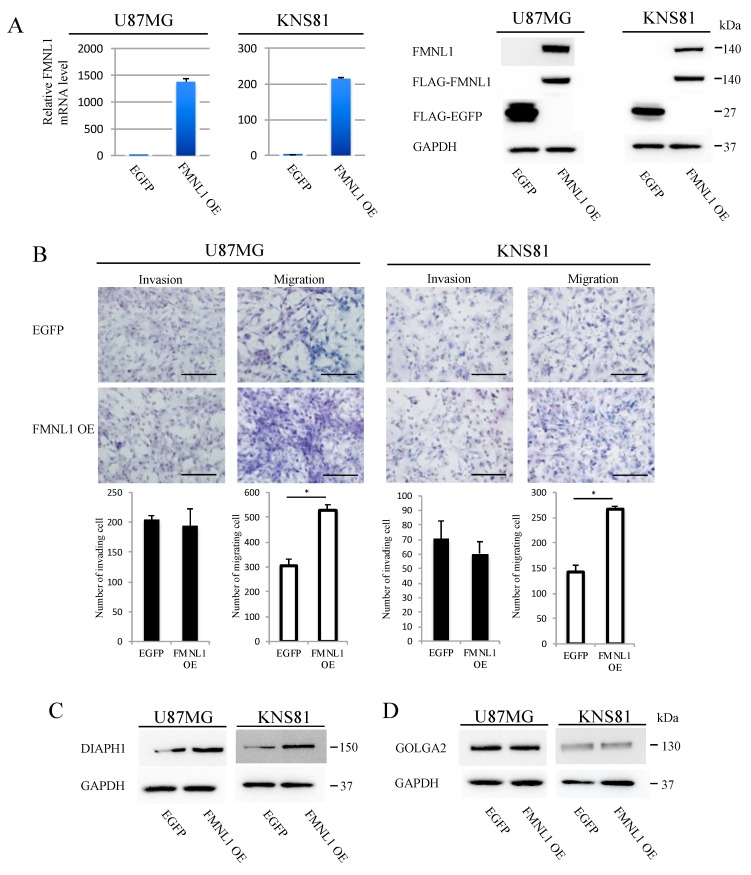
FMNL1 overexpression induces migration but not invasion. (**A**) Reverse transcription (RT)-PCR (left) and Western blot (right) of FMNL1 in U87MG and KNS81 cells infected with FMNL1 or EGFP lentiviral expression vectors. (**B**) Representative images of transwell migration and invasion of cells overexpressing FMNL1 and EGFP shown at 200× magnification, which were quantified. Scale bar: 100 μm. Columns represent migrating or invading cells from five independent microscopic fields, with bars indicating SD. * *p* < 0.01 vs. cells overexpressing EGFP. (**C**,**D**) Protein expression of DIAPH1 and GOLGA2 in (**C**) U87MG and (**D**) KNS81 cells transfected with FMNL1 or EGFP lentiviral expression vectors.

**Figure 5 ijms-20-06355-f005:**
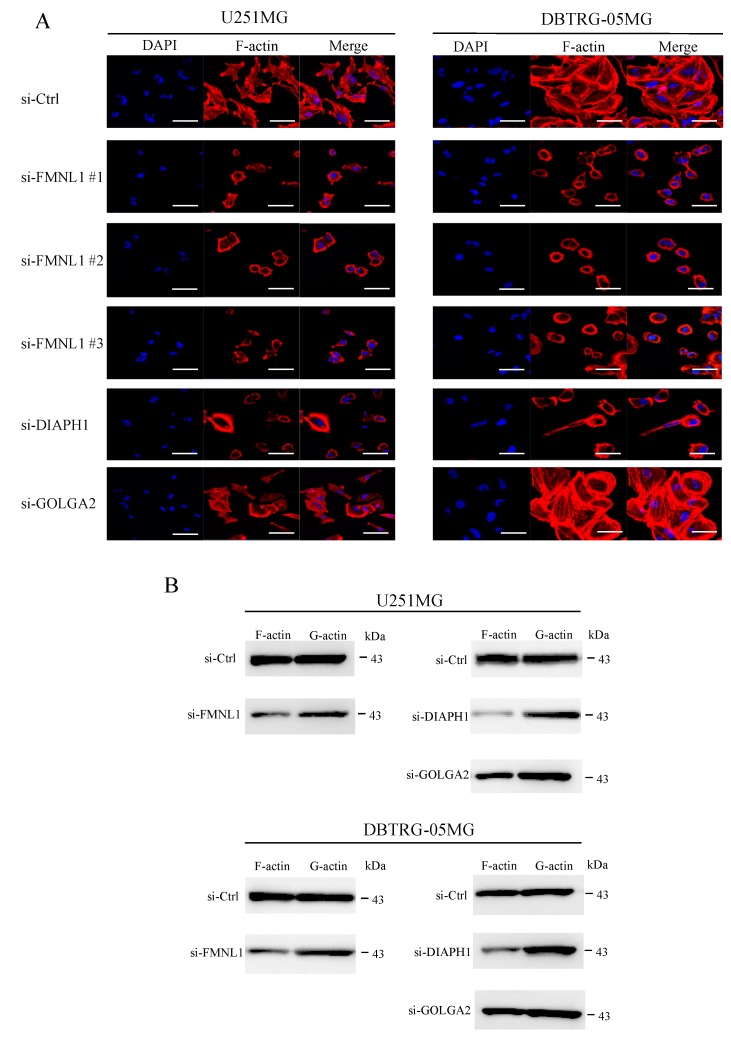
Knockdown of FMNL1 compromises the actin cytoskeleton in GBM cells via DIAPH1 but not GOLGA2. (**A**) U251MG and DBTRG-05MG cells were transfected with control siRNA or siRNAs against FMNL1, DIAPH1, or GOLGA2, and stained with DAPI and ActinRed^TM^ 555 ReadyProbes^TM^. Original magnification 400×; scale bar: 50 μm. (**B**) F (filamentous) and G (globular) fractions of actin in cell lysates were assessed by Western blot using a G-actin/F-actin in vivo assay kit.

**Table 1 ijms-20-06355-t001:** Univariate and multivariate Cox regression analysis of overall survival in GBM patients.

	Univariate Analysis			Multivariate Analysis		
	HR	95% CI	*p*-Value	HR	95% CI	*p*-Value
Sex	1.081	0.82–1.43	0.589	0.957	0.71–1.29	0.773
Age	1.008	1.01–1.03	< 0.001	1.013	1.00–1.02	0.009
KPS (> 70)	0.522	0.39–0.70	< 0.001	0.749	0.55–1.03	0.072
Ki-67 (> 30%)	0.986	0.74–1.31	0.919	0.794	0.59–1.07	0.132
Extent of surgical resection (Total + Subtotal/Partial + Biopsy)	0.466	0.35–0.62	< 0.001	0.631	0.46–0.86	0.004
Number of surgeries (Single/Multiple)	1.685	1.14–2.50	0.009	1.329	0.85–2.07	0.208
Chemotherapy (Yes/No)	0.202	0.13–0.31	< 0.001	0.356	0.20–0.64	< 0.001
Bevacizumab (Yes/No)	0.716	0.51–1.00	0.051	0.743	0.53–1.04	0.087
Radiation therapy	0.979	0.97–0.98	< 0.001	0.987	0.98–0.99	< 0.001
FMNL1 expression (High/Low)	0.534	0.40–0.71	< 0.001	0.631	0.46–0.86	< 0.001

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
