# Peer review of "Formin-like 1 (FMNL1) Is Associated with Glioblastoma Multiforme Mesenchymal Subtype and Independently Predicts Poor Prognosis"

_ijms, 2019, doi:10.3390/ijms20246355_

Round 1

Reviewer 1 Report

In this manuscript, the authors explored the potential roles of formin-like 1 (FMNL1) in glioblastoma (GBM). They demonstrated that expression of FMNL1 inversely correlates with GBM patient survival, and was higher in the mesenchymal subtype. They revealed that FMNL1 regulated GBM cell migration and invasion via DIAPH1 and GOLGA2, repectively. Furthermore, FMNL1 downregulation compromised the actin cytoskeleton via DIAPH1 but not GOLGA2. Overall, this is an interesting story.

There are several issues which need to be addressed of fixed.

Have the authors compared the expression of FMNL1 between in GBM tumor tissues and normal brain tissues by database analysis? In Figure 1D, the patients’ number of FMNL1 high/FMNL1 low should be presented in the graph. Figure 1F looks fuzzy. The picture quality needs to be improved. Is the order of Line 138 and 139 inverse in Figure 2 Legend? Can authors provide shorter-exposed western blot images for DIAPH1 in DBTRG-05MG in Figure 3A and GOLGA2 in U251MG in Figure 3D? How were the proteins loaded for western blot in Figure 5B? Usually the level of F-actin is not equal to that of G-actin in cells.

Author Response

Response> We appreciate your kind comments and useful suggestions.

Have the authors compared the expression of FMNL1 between in GBM tumor tissues and normal brain tissues by database analysis?

Response > We didn’t compare FMNL1 expression in the database, because we could not detect FMNL1 expression immunehistochmically and previously it has been reported that no expression was detected iin normal brain reference 26.

In Figure 1D, the patients’ number of FMNL1 high/FMNL1 low should be presented in the graph.

Response>We appreciate your suggestion. We added the patients’ number in the graph Figure 1D.

Figure 1F looks fuzzy.

Response>We appreciate your suggestion. GSEA application of Broad institute and there is a limitation of quality.

Is the order of Line 138 and 139 inverse in Figure 2 Legend?

Response>Thank you for your suggestion. We change the sentence in figure2 legend. We hope it is acceptable for you as follow “FMNL1 knockdown suppressed MMP9 activity; MMP9 activity in the cell-conditioned medium was evaluated by gelatin zymography (E).”

Can authors provide shorter-exposed western blot images for DIAPH1 in DBTRG-05MG in Figure 3A and GOLGA2 in U251MG in Figure 3D?

Response> I am sorry we don’t have shorter expose data in comparison with the picture.

How were the proteins loaded for western blot in Figure 5B? Usually the level of F-actin is not equal to that of G-actin in cells. 

Response> We appreciate the reviewer’s comments.  We used G-actin/F-actin in vivo assay kit. In the experiment F and G- actin were collected from one fraction of cells.  The status of actin polymerization can be fixed with the kit buffer and we could separate F and G- actin by molecular weight using centrifuge.  In SDS PAGE buffer F-actin was depolymerised and resolved, so we cannot distinguish between G-action and F-action in the condition.  In Fig. 5, we compared the change of ratio of F and G- actin by knockdown of each gene.   

We hope we could improve that in the new manuscript, although the image supplied by

Reviewer 2 Report

The manuscript “Formin-like 1 (FMNL1) is associated with glioblastoma multiforme mesenchymal subtype and independently predicts poor prognosis” concerns the potential role of FMNL1 in regulation of GBM invasive properties. Although the hypothesis is clear some experiments seems to be executed not in a proper way. Unfortunately, the presented results and the structure of the text are not convincing enough to support the thesis. The work requires a thorough review:

methodology raises my biggest concern. Authors use the transwell assay (modified Boyden chambers) to investigate both the cells invasiveness and motility. Although the description of the method is not clear I suppose that the only difference is the existence of additional barrier (fibrinogen etc.). Actually, it doesn’t matter. The transwell assay is not a sufficient method for the cell motility analyses. It can be used for the invasive assay and often the “naked” transwell membrane is covered with ECM, Matrigel or endothelial cells to reflect the natural barriers in metastasis process. Additionally authors fill the bottom chamber with the medium containing FBS (upper chamber without FBS). This modification strictly change the test into chemotactic one. It is a methodological mistake because all the cells migrate into FBS and situation with two compartments with/without FBS does not exist in a living organism. Other words, the authors force the cells to move if they want to survive. In my opinion it is not possible to use the transwell assay in both invasion and motility investigation. For the cell motility examination the best way is the time-lapse video microscopy with further analysis enabling to obtain cell speed, displacement, total trajectory length etc. These two (transwell and time-lapse) methods are often use in recent works and give complementary and more sufficient results (see Pudełek, 2019, Fitoterapia 134, 172-181). I suggest the authors to use more proper method for movement analysis. I have doubts about the cell culture. In the manuscript RPMI 1640 is described as a culture media. Most of GBM cell lines is cultured in EMEM or DMEM with high glucose. I checked the recommendation at ATCC and similar sources, and only the DBTRG-05MG cell line should be cultured in RPMI medium but supplemented with L-proline, L-cystine, HEPES, hypoxanthine, adenosine triphosphate etc. Moreover I had a problem to find the U251MG and KNS81 cells in Japanese Collection of Research Bioresources Cell Bank. Using of not dedicated media for the cell culture may impact the results and make them not repeatable in other laboratories. I found in figure 3 that the bands for GAPDH in A (for U251MG) and D (for DBTRG-05MG) are identical. Moreover they are also identical in supplementary images of WB. It rises my doubts about the accuracy of the data, I’m not sure if I can thrust the rest of the results.

Minor problems

“Cytoskeletal remodeling processes such as cell polarization, adhesion, division, and migration are strongly dependent on formin proteins function” - this sentence is misleading. Remodeling is not a adhesion and so on… Use the bars in all pictures to show the scale. The information about magnification is not sufficient Specify the number of wells in multi-well plate used for MTT. MTT is not mentioned in manuscript text (I can’t open the supplementary figures, maybe it is there) text should be checked by native speaker

Author Response

Response about the method> We appreciate the reviewer’s useful comments.  I agree there are several better methods to evaluate motility and invasiveness more precisely. However transwell methods are most common and widely used in cancer migration and invasion estimation.  Many researchers use FCS as attractant for invasion assay to induce the cells enter to matrigel in previous papers. I attach the list of papers using the similar invasion assay. 

Kogo, Ryunosuke et al. 2011. “Long Noncoding RNA HOTAIR Regulates Polycomb-Dependent Chromatin Modification and Is Associated with Poor Prognosis in Colorectal Cancers.” Cancer Research 71(20): 6320–26. Nishita, Michiru et al. 2017. “Ror2 Signaling Regulates Golgi Structure and Transport through IFT20 for Tumor Invasiveness.” Scientific Reports 7(1): 1–15. Wu, Yanxia et al. 2017. “High FMNL3 Expression Promotes Nasopharyngeal Carcinoma Cell Metastasis: Role in TGF-Β1-Induced Epithelia-to-Mesenchymal Transition.” Scientific Reports 7(September 2016): 1–14. Zóia, Mariana Alves Pereira et al. 2019. “Inhibition of Triple-Negative Breast Cancer Cell Aggressiveness by Cathepsin D Blockage: Role of Annexin A1.” International Journal of Molecular Sciences 20(6). Fang, Li Li et al. 2017. “Potent Inhibition of MiR-34b on Migration and Invasion in Metastatic Prostate Cancer Cells by Regulating the TGF-β Pathway.” International Journal of Molecular Sciences 18(12): 1–17. Wang, Xuanbin et al. 2016. “Up-Regulation of PAI-1 and down-Regulation of UPA Are Involved in Suppression of Invasiveness and Motility of Hepatocellular Carcinoma Cells by a Natural Compound Berberine.” International Journal of Molecular Sciences 17(4): 1–15.

Response about culture condition> We appreciate the reviewer’s comments and understand the reviewer’s concern. We used RPMI because we followed several previous papers in the list below.  We didn’t have any problem in cell proliferation and our condition is not complicated so we believe the other researchers can reproduce same result in the same conditions. I attach the reference list below.

DBTRG-05MG

Tsai N et al. The antitumor effects of angelica senensis on malignant brain tumors in vitro and in vivo. Clin Cancer Res 2005; 11(9) Tsai N et al. The natural compound n-butylidenephthalide derived from angelica senesis inhibits malignant brain tumors in vitro and in vivo. Journal of Neurochemistry 2006, 99, 1251-1262 Ferruzzi P et al. In vitro and vivo characterization of a novel hedgehog signaling antagonist in human glioblastoma cell lines. Int. J. Cancer 2012, 131, E33-E44 Harn H et al. Local interstitial delivery of z-butylidenephthalide by polymer wafers against malignant human gliomas. Neuro-Oncology 2011, 13(6):635-648 Filho PC et al. CdTe quantum dots as fluorescent probes to study transferrin receptors in glioblastoma cells. Biochimica et Biophysica Acta 2016, 1860;28-35

U251MG

Prabhu A et al. Cysteine catabolism: a novel metabolic pathway contributing to glioblastoma growth. Cancer Res 2014, 74(3) Musah-Eroje A et al. Adaptive changes of glioblastoma cells following exposure to hypoxic (1% oxygen) tumor microenvironment. Int. J. Mol. Sci 2019, 20,2019

U87MG

Zappavigna S et al. A hydroquinone-based derivative elicits apotosis and autophagy via activating a ROS-dependent unfolded protein response in human glioblastoma. Int. J. Mol. Sci 2019, 20, 3836 Wen YT et al. A novel multi-target small molecule, LCC-09, inhibits stemness and therapy-resistant phenotypes of glioblastoma cells by increasing miR-34a and Deregulating the DRD4/Akt/mTOR signaling axis. Cancers 2019, 11, 1442 Ru Q et al. Voltage-gated potassium channel blocker 4-aminopyridine induces glioma cell apoptosis by reducing expression of microRNA-10b-5p. Molecular Biology of the Cell 2018, volume 28 Liu HW et al. The disruption of the β-Catenin/TCF-1/STAT3 signaling axis by 4-acetylantroquinonol B inhibits the tumorigenesis and cancer stem-cell-like properties of glioblastoma cells, in vitro and in vivo. Cancers 2018, 10, 491

Response about GAPDH image>>We really apologize this mistake. We have replaced  a correct photo for Fig. 3D GAPDH. It accidentally happened.  I swear to god this is just a simple mistake.

Response about remodeling>We agree the authors comment.  We change the sentence as follow “Cytoskeletal remodeling processes which related to cell polarization, adhesion, division, and migration are strongly dependent on formin proteins function” 

Response about scale bars>We appreciate the reviewer’s comment.  We added scale bars in the pictures.

Response about magnificent>> We appreciate the reviewer’s comment. We added magnificent information to the figure legends.

Response about plate>>Thank you for your comment. We added “96 well plates” in the Cell proliferation assay section of materials and methods.

Response about MTT >MTT is (3-(4,5-dimethylthiazol-2-yl)-2,5-diphenyl tetrazolium bromide is described in the reagents and antibody section.  I am sorry we had a problem supplementary files at first.  I believe you can see them now.  Actually the text were edited by a native English speaker.  

Reviewer 3 Report

In this article entitled “Formin-like 1 (FMNL1) is associated with glioblastoma multiforme mesenchymal subtype and independently predicts poor prognosis” by Nayuta Higa, et al. described about essential roles of FMNL1, whose expression was positively associated with poor prognosis and mesenchymal subtype in the series of GBM cases from author’s institute, in regulation of migration and invasion via DIAPH1 and GOLGA2, respectively. In this study, the authors used several commercial human GBM cell lines and investigated by in vitro assay.

# Comments:

The molecular weight size should be depicted in every immunoblot analysis.

The images of microscope lack scale bars. These should be added.

The authors performed migration/invasion assay of the cells after 24 hours from passage; however, the time point is too late for the assay (usually, after 6~8 hours from passage) because the effect of proliferation cannot be ignored. The data of invasion assay should be obtained earlier time points or corrected considering the proliferation rate.

It is more preferable to demonstrate graphical abstract or summary.

In all siRNA-mediated knockdown experiments, at least two siRNAs with different sequence should be used against one target molecule.

In all gene overexpression experiments, the data should be salvaged by further knockdown of overexpressed gene.

The authors should explain the reason why they used different GBM cell lines only in figure 4.

In figure 4A, the data of immunoblotting using anti-FMLN1 antibody is also required.

If GOLGA2 regulates GBM cell invasion but not migration, it is also necessary to demonstrate whether GOLGA2 is involved in regulation of GBM cells-mediated extracellular matrix degradation by such as gelatin zymogram assay.

Author Response

Response about molecular weight> We agree with the reviewer’s comment. We added molecular weight size marks in the immunoblotting data. 

Response about scale bars> We appreciate the reviewer’s comment. We added sale bars in the microscope photographs.  

Response about proliferation > We appreciate the reviewer’s comments.  FMNL1 expression didn’t affect to cell proliferation in 24 hours in supplementary figure 4 and describe as in text as “FMNL1 effect knockdown on cell proliferation was not significant during the term in invasion and migration assays (Supplementary Fig. S4)”. We judged that we could ignore the effect of cell proliferation on cell numbers of migration and invasion assay at the time point.  The timing of the assays depend on the conditions and cells.  To observe clear difference migration and invasion of the cells, we judged it was proper to estimate cell number 24 hours after passage.  

Response about graphic abstract > We had added a graphical abstract, however It didn’t reflect well. I asked that the editorial staff can help us to appear that.    

Response about siRNA> We appreciate and understand the reviewer’s comments.  We used the siRNA commercially evaluated siRNA here. We confirmed that GOLGA2 siRNA was used in BBA general subjects 1891, (2017) p2891-2901.

 Response about salvage experiments> We appreciate the reviewer’s comments. We agree that it is better to add siRNA on FMNL1 overexpressing cells. However we ascertain that siRNA FMNL1 can clearly affected migration and invasion endogenously FMNL1 expressing cells (in FIG.2). It is clear that FMNL1 expression is relate to migration and invasion. 

Response about cells in FIG.4 > We appreciate the reviewer’s comments.  These two cell lines (U87MG and KNS81) slightly expressed FMNL1 and we judge that they are proper to use for overexpression experiments in figure. 4. U251-MG and DBTRG-05MG cells express more FMNL1 endogenously so we used them for knockdown experiments. 

Response about Anti-FMNL1 antibody data in FIG.4 > We appreciate and agree the reviewer’s comment. We add immunblotting data of anti-FMNL1 in FIG.4A.  

Response about Golga2 function > We appreciate the reviewer’s comments.  We would like to perform the experiment as future studies with further detail analysis of GOlGO2 function, since we have to examine several experiments about invasion relating molecules including MMPs.   

Round 2

Reviewer 2 Report

The authors should know that the RPMI medium contains half of dose of phenol red in comparison to dmem, emem etc. Phenol red is a quite active xeonoestrogen which causes a differences in cell movement when you use RPMI and other media. My problem is not the ability to repeat the results described in manuscript but to compare them with other results obtained by other groups using proper media. GBM cells are able to react to xenoestrogens by ERy receptors so the change of media is not suggested. And citation of other works using RPMI for GBM cell culture is not an argument. They are all wrong. However it would be not reasonable to reject the manuscript or suggest to repeat all experiments with other culture media. Authors should concern this problem in future experiments.

Reviewer 3 Report

Response about molecular weight> We agree with the reviewer’s comment. We added molecular weight size marks in the immunoblotting data.

⇒ I satisfied with author’s responses.

Response about scale bars> We appreciate the reviewer’s comment. We added sale bars in the microscope photographs.

⇒ I confirmed the change of figure 1, however, scale bars should be also added to figure 2C,figre 2D, figure 3C, figure 3F, figure 4B, and figure 5A.

Response about proliferation > We appreciate the reviewer’s comments. FMNL1 expression didn’t affect to cell proliferation in 24 hours in supplementary figure 4 and describe as in text as “FMNL1 effect knockdown on cell proliferation was not significant during the term in invasion and migration assays (Supplementary Fig. S4)”. We judged that we could ignore the effect of cell proliferation on cell numbers of migration and invasion assay at the time point. The timing of the assays depend on the conditions and cells. To observe clear difference migration and invasion of the cells, we judged it was proper to estimate cell number 24 hours after passage.

⇒ I confirmed the data of supplementary figure 4 and considered proliferation of the cells would not affect the results of migration assay. Therefore, it is OK.

Response about graphic abstract > We had added a graphical abstract, however It didn’t reflect well. I asked that the editorial staff can help us to appear that.

⇒ It is OK.

Response about siRNA> We appreciate and understand the reviewer’s comments. We used the siRNA commercially evaluated siRNA here. We confirmed that GOLGA2 siRNA was used in BBA general subjects 1891, (2017) p2891-2901.

⇒ The most important point of my indication was not efficiency but specificity of each siRNA. Therefore, it is necessary to try multiple siRNAs with different sequences against one target event in the cases of validated commercial siRNAs. In this report, all of the results of knockdown experiments with single siRNA were clear, therefore, I consider these results would be correct. However, please endeavor my suggestion henceforward.

Response about salvage experiments> We appreciate the reviewer’s comments. We agree that it is better to add siRNA on FMNL1 overexpressing cells. However we ascertain that siRNA FMNL1 can clearly affected migration and invasion endogenously FMNL1 expressing cells (in FIG.2). It is clear that FMNL1 expression is relate to migration and invasion.

⇒ Like as my previous question, please keep my suggestion in mind. Because the results regarding to my questions were obvious, reluctantly, I will settle this for now.

Response about cells in FIG.4 > We appreciate the reviewer’s comments. These two cell lines (U87MG and KNS81) slightly expressed FMNL1 and we judge that they are proper to use for overexpression experiments in figure. 4. U251-MG and DBTRG-05MG cells express more FMNL1 endogenously so we used them for knockdown experiments.

⇒ I accepted author’s reply. Then, the authors should mention about that in “Results” section.

Response about Anti-FMNL1 antibody data in FIG.4 > We appreciate and agree the reviewer’s comment. We add immunblotting data of anti-FMNL1 in FIG.4A.

⇒  I confirmed the change of figure 4A. It is OK.

Response about Golga2 function > We appreciate the reviewer’s comments. We would like to perform the experiment as future studies with further detail analysis of GOlGO2 function, s

⇒ I acknowledged author’s reply. Then, the authors should mention about that in “Discussion” section.